# Reciprocal Reward Influence Encourages Cooperation From Self-Interested Agents

**John L. Zhou**    **Weizhe Hong**    **Jonathan C. Kao**
University of California, Los Angeles
john.ly.zhou@gmail.com

## Abstract

Cooperation between self-interested individuals is a widespread phenomenon in the natural world, but remains elusive in interactions between artificially intelligent agents. Instead, naïve reinforcement learning algorithms typically converge to Pareto-dominated outcomes in even the simplest of social dilemmas. An emerging literature on opponent shaping has demonstrated the ability to reach prosocial outcomes by influencing the learning of other agents. However, such methods differentiate through the learning step of other agents or optimize for meta-game dynamics, which rely on privileged access to opponents' learning algorithms or exponential sample complexity, respectively. To provide a learning rule-agnostic and sample-efficient alternative, we introduce Reciprocators, reinforcement learning agents which are intrinsically motivated to reciprocate the influence of opponents' actions on their returns. This approach seeks to modify other agents' $Q$-values by increasing their return following beneficial actions (with respect to the Reciprocator) and decreasing it after detrimental actions, guiding them towards mutually beneficial actions without directly differentiating through a model of their policy. We show that Reciprocators can be used to promote cooperation in temporally extended social dilemmas during simultaneous learning. Our code is available at https://github.com/johnlyzhou/reciprocator/.

## 1   Introduction

Many species exhibit cooperative behaviors of remarkable variety and complexity. Even among prosocial animals, however, humans are especially notable for their ability to cooperate with unrelated individuals and maintain that cooperation even in highly adversarial environments (Melis & Semmann, 2010). These qualities are often credited with the development of human technology, culture, and advanced cognition (Burkart et al., 2014). As artificially intelligent (AI) agents become more commonplace in human society, it is increasingly important to ensure that they share similar prosocial qualities so that interactions between AI agents, as well as between AI agents and humans, may converge to mutually beneficial outcomes.

However, state-of-the-art reinforcement learning (RL) methods are typically designed for the single agent setting and are ill-suited for the nonstationarities introduced by multiple agents learning simultaneously. Treating other agents as fixed elements of the environment, referred to as the "naïve" learning (NL) approach, can destabilize training and produce collectively suboptimal outcomes. This is particularly evident in a class of games known as sequential social dilemmas (SSDs), which contain tradeoffs between collective and individual return (Leibo et al., 2017). SSDs present a particularly challenging problem for multi-agent RL (MARL) because of this mixed motivational structure, which precludes the use of common centralized training algorithms designed for cooperative settings (Kraemer & Banerjee, 2016; Sunehag et al., 2017; Gupta et al., 2017; Rashid et al., 2018).

38th Conference on Neural Information Processing Systems (NeurIPS 2024).

Previous work has shown that NL agents optimizing only for their individual returns converge to non-cooperative, Pareto-dominated outcomes in even the simplest of SSDs (Foerster et al., 2018).

In this work, we propose an intrinsic reward that encourages agents called *Reciprocators* to learn a tit-for-tat-like strategy which reciprocates the influence that others exert on their returns. We first define *value influence*, a notion of influence that quantifies the effect that one agent's action has on another's expected return. Given a pair of agents, a Reciprocator $rc$ and another learning agent $i$, we track the cumulative influence of agent $i$'s sequence of actions on $rc$'s expected return, which we refer to as the *influence balance* "owed" to $rc$ by agent $i$. At each time step, the Reciprocator receives an additional intrinsic *reciprocal reward* proportional to the product of its current influence balance with agent $i$ and the value influence of its action on agent $i$. This encourages the Reciprocator to take actions whose influence matches the sign of the influence balance: for example, if agent $i$ has produced a net positive influence on $rc$, then $rc$ will be rewarded for actions that have a reciprocally positive influence on agent $i$'s expected return. Mechanistically, our method seeks to manipulate the opponent's $Q$-values by altering their expected return in particular directions depending on their actions. A reward-maximizing agent $i$ should then be incentivized to take mutually beneficial actions and avoid harmful externalities.

The contributions of this work are as follows: (1) We formulate a novel intrinsic reward that encourages an agent to incentivize cooperation from other, *simultaneously learning* agents without modifying the structure of the environment, computing higher-order derivatives, or learning meta-game dynamics. (2) Agents trained with this reward achieve state-of-the-art cooperative outcomes in sequential social dilemmas and are able to shape purely self-interested naïve learners into mutually beneficial behavior. (3) Reciprocators demonstrate resistance to exploitation by higher-order baselines, despite using only *first-order* reinforcement learning algorithms.

## 2 Related Work

In order to improve convergence towards Pareto-optimal solutions among independently learning agents, previous work has modified agents' reward structures to explicitly consider either group or per-capita return in order to promote cooperative behavior, e.g., via inequity aversion (Hughes et al., 2018), "empathic" harm reduction (Bussmann et al., 2019), or "altruistic" gradient adjustments (Li et al., 2024). While agents that abide by such restrictions may seek to reduce their harmful externalities, they have no way of regulating the behavior of other, less magnanimous agents, including purely self-interested "exploiters" in the worst case (Agapiou et al., 2023). Their efficacy in SSDs, as well as in most other types of multi-agent systems, is therefore limited in the absence of strong guarantees over the other agents' altruistic tendencies.

### 2.1 Cooperation through Influence

A more robust form of cooperation can be achieved by actively exerting influence over other agents in the environment rather than unilaterally adopting a prosocial policy, especially in SSDs where extrinsic rewards incentivize defection. Jaques et al. (2019) provides an intrinsic reward for "social influence" on other agents, which is computed as the Kullback-Leibler (KL) divergence between an opponent's policy distribution conditioned on an *influence agent*'s action $a^i$ and a counterfactual distribution which marginalizes out that influence agent. However, as a metric-agnostic quantity, KL divergence is unable to capture key details of *how* the action distribution changes (Park et al., 2024). In particular, this definition of influence cannot selectively modify opponents' behavior with respect to extrinsic task returns or state transition dynamics, and therefore has limited utility as a mechanism to incentivize cooperation.

Other works seek to directly influence opponent returns via the social mechanisms of reward and punishment, which are believed to assist in stabilizing cooperative relationships by controlling free-riding, cheating, and other antisocial behaviors (Melis & Semmann, 2010). Incorporating these mechanisms into groups of agents, whether by providing rewards to incentivize good behavior (Yang et al., 2020), or doling out punishments to discourage bad behavior (Schmid et al., 2021; Yaman et al., 2023), has been shown to encourage cooperation among independent agents in a variety of SSDs. However, these methods require extensions of the original action space that allow agents to directly modify other agents' rewards. On the other hand, we instead propose to quantify the influence of each naturally available action $a \in A$ in a given state $s$ on another agent's expected return $R$, and use

this influence as the medium for reciprocation. As we later show, selectively rewarding or punishing an opponent's actions can be seen as a form of *opponent shaping* which seeks to manipulate the $Q$-value of given state-action pair in order to influence the likelihood of that action in future policies.

## 2.2 Opponent Shaping

Considering future policies points to a key issue with the canonical reinforcement learning (RL) framework, in that it optimizes only for the expected return within a single episode of the environment. However, taking actions that seek to optimize the long-term behavior of the other agents in the environment will not receive any immediate feedback within an episode, and must wait for one, or possibly many, learning steps. Opponent-shaping methods of this kind address this issue by differentiating across pairs of episodes (Yang et al., 2020), through opponents' gradient updates (Foerster et al., 2018; Zhao et al., 2022; Willi et al., 2022), or repeated sequences of learning steps organized into "meta-episodes" (Lu et al., 2022). While these methods have demonstrated convergence to cooperative behavior in simple SSDs, they are impractical or intractable in realistic multi-agent scenarios, requiring additional independent action channels for providing incentives, white-box access to the learning rules and gradients of other agents, or exponential sample complexity (Fung et al., 2023) to learn the dynamics of meta-games, respectively.

In particular, the Learning with Opponent-Learning Awareness (Foerster et al., 2018, LOLA) class of approaches differentiate through the opponent's learning update but have only been demonstrated to work with full access to either the opponent's policy gradients and Hessians or their policy parameters. When using gradient approximations and modeling the opponent's policy, LOLA with opponent modeling (LOLA-OM) showed significantly worse results, even against vanilla policy gradients. Modeling efforts become even more implausible when faced with modern RL techniques, which employ adaptive optimizers that set different per-weight learning rates (Kingma & Ba, 2017, Adam), randomized experience replay (Schaul et al., 2016), and various auxiliary terms such as policy divergence penalties (Schulman et al., 2017) and entropy exploration bonuses (Williams, 1992; Mnih et al., 2016) to improve learning. Model-Free Opponent Shaping (Lu et al., 2022, MFOS) was introduced as a more general meta-learning approach that requires neither privileged access to nor inconsistent assumptions on opponent learning rules. However, MFOS's meta-policies are trained across multiple training runs repeated in sequence, each of which are treated as a single "meta-episode," in order to learn how to exploit opponent learning dynamics. This requires the ability to freely perform rollouts in the environment against opponents whose policies can be repeatedly reset to initialization, an unrealistic assumption in environments with partially adversarial motivations.

Our work is conceptually most similar to Learning with Opponent Q-Learning Awareness (Aghajohari et al., 2024, LOQA), which also seeks to shape opponent policies by influencing the $Q$-values for different actions, under the assumption that opponents are Q-learners. However, LOQA still differentiates through a model of the opponent's policy and optimizes according to the joint advantage function $A(s_t, a_1, a_2)$ computed with respect to the state-value function $V(s_t)$. On the other hand, we use a counterfactual baseline that marginalizes out only the opponent's action to perform *agent-specific* credit assignment and use an intrinsic reward instead of gradients over opponent policies to encourage agents to influence the opponent's $Q$-values in the correct direction. Most importantly, LOQA focuses on the problem of learning a general end-policy that performs well against a variety of other agents *at evaluation* and has therefore only demonstrated the ability to shape other LOQA opponents in a controlled self-play scenario, while our method focuses on shaping the policies of diverse agents over multiple episodes of *simultaneous learning*.

In the context of these higher-order opponent-shaping approaches, we position our intrinsic reciprocal reward as a form of immediate, within-episode feedback that encourages otherwise naïve learners to implicitly consider the long-term, cross-episode effects of their actions on the policy changes of other agents, demonstrating many of the properties of higher-order opponent-shaping methods while using only first-order reinforcement learning algorithms and standard rollout-based training procedures.

## 3 Preliminaries

We construct a series of sequential social dilemmas which can each be described as a stochastic game $G$, defined by a tuple $G = \langle S, A, P, r, n, \gamma \rangle$. In $G$, $n$ agents, indexed by $i \in \{1, \ldots, n\}$, observe the state of the environment $s \in S$ and simultaneously choose actions $a^i \in A$ to form a joint action

$\boldsymbol{a} \in \boldsymbol{A} \equiv A^n$. The environment then undergoes a change according to the state transition function $P(s'|s, \boldsymbol{a}) : S \times \boldsymbol{A} \times S \to [0, 1]$. Each agent receives an individual reward $r^i = r^i(s, \boldsymbol{a})$, and future rewards are discounted at each time step by a discount factor $\gamma \in [0, 1]$. We consider the fully observable setting where agents have access to the full state of the environment $s \in S$ at every time step, joint action $\boldsymbol{a} \in \boldsymbol{A}$, and rewards received by each agent. Each agent $i$ conditions a stochastic recurrent policy $\pi^i(a^i|\tau^i)$ on its action-observation history, which is denoted as $\tau^i \in T \equiv (S \times \boldsymbol{A})^*$.

## 4 Reciprocal Reward Influence

### 4.1 1-Step Value Influence

In order to compute a general measure of value-based influence, we modify an intrinsic exploration reward proposed by Wang et al. (2020) called Value of Interaction (VoI). VoI measures how much agent $i$'s actions affect agent $j$'s expected return by computing the difference between the $Q$-value conditioned on the joint state and action $Q_j^{\boldsymbol{\pi}}(s, \boldsymbol{a})$ and a counterfactual baseline $Q_{j|i}^{\boldsymbol{\pi}}(s, \boldsymbol{a}^{-i})$ which marginalizes out the state and action of the influencing agent $i$ to compute a "default" expected return [Equation 1]. Here, the bolded $\boldsymbol{\pi}$ superscript indicates the dependence of these $Q$-functions on the parameters of all agents' policies. The VoI can be decomposed into an immediate influence term $r(s, \boldsymbol{a}) - r(s, \boldsymbol{a}^{-i})$ and a discounted future influence term computed over changes in state transition probabilities [Equation 2]. Although the original VoI was formulated as an expectation over trajectories $\tau$ and used as a regularizer, we modify it to compute the one-step Value Influence $(VI)$ where $VoI = \mathbb{E}_\tau[VI]$, allowing us to quantify the influence of individual actions

$$VI_{i|j}^{\boldsymbol{\pi}}(s, \boldsymbol{a}) = Q_j^{\boldsymbol{\pi}}(s, \boldsymbol{a}) - Q_{j|i}^{\boldsymbol{\pi}}(s, \boldsymbol{a}^{-i}) \tag{1}$$

$$= r(s, \boldsymbol{a}) - r(s, \boldsymbol{a}^{-i}) + \gamma \sum_{s'} \left(1 - \frac{p^{\pi^i}(s' \mid s, \boldsymbol{a}^{-i})}{p(s' \mid s, \boldsymbol{a})}\right) V(s'), \tag{2}$$

where $r(s, \mathbf{a}^{-i}) = \mathbb{E}_{a^i \sim \pi^i}\left[r(s, (\mathbf{a}^{-i}, a^i))\right]$ and $p(s'|s, \mathbf{a}^{-i}) = \mathbb{E}_{a^i \sim \pi^i}\left[p^{\pi^i}(s'|s, (\mathbf{a}^{-i}, a^i))\right]$. This definition of influence relies on the notion of a counterfactual baseline to assign credit to a particular agent's action while holding all other agents' actions constant. This counterfactual baseline is so named because it estimates the counterfactual expected return if the agent's true action is replaced with a "default" action, which is computed by marginalizing out the agent's action to get the expected on-policy return. This baseline return is given by

$$Q_{j|i}^{\boldsymbol{\pi}}(s, \boldsymbol{a}^{-i}) = \mathbb{E}_{a^i \sim \pi^i}\left[Q_j^{\boldsymbol{\pi}}(s, (\boldsymbol{a}^{-i}, a^i))\right] = \sum_{a_i} \pi^i(a_i|\tau^i) Q_j^{\boldsymbol{\pi}}(s, (\boldsymbol{a}^{-i}, a^i)). \tag{3}$$

In practice, we approximate this by regressing towards the $Q$-value while masking out $a_i$ from the joint state-action input. Marginalizing out an agent's action to quantify influence is a common paradigm in MARL, having also been used to define 1-step adversarial power (Li & Dennis, 2023), which estimates the maximum reduction in agent $j$'s expected reward that can be achieved by agent $i$ as the minimum $VI_{i|j}^{\boldsymbol{\pi}}(s, \boldsymbol{a})$ over all $a^i \in A$. Similarly, Counterfactual Multi-Agent policy gradients (Foerster et al., 2017, COMA) marginalizes out a single agent's action to assess the advantage (i.e., influence) of that agent's selected action relative to the counterfactual on-policy return *ceteris paribus*. Note that the COMA advantage function [Equation 4] can be expressed a special case of $VI_{i|j}^{\boldsymbol{\pi}}(s, \boldsymbol{a})$ where $i = j$.

$$A_i(s, \mathbf{a}) = Q_i(s, \mathbf{a}) - \sum_{a_i} \pi^i(a_i|\tau^i) Q_i^{\boldsymbol{\pi}}(s, (\boldsymbol{a}^{-i}, a^i)). \tag{4}$$

### 4.2 Keeping Score with Influence Balances

The amount of influence that a Reciprocator is able to exert in a single time step is heavily environment-dependent, so that it may not always be possible to immediately reciprocate previous influences. To encourage reciprocation over extended timescales, we continuously accumulate a measure of net

---

**Algorithm 1** Training with Reciprocal Reward Influence vs. Agent $i$

---

Initialize agent $rc$'s policy parameters $\theta_\pi$, VI target function parameters $\phi$, influence balance vector $B_{rc|i} = \mathbf{0}$, policy memory $\mathcal{M} = \emptyset$, and influence memory $\mathcal{H} = \emptyset$
**for** each episode $e$ **do**
    Observe initial state $s_0$
    **for** $t = 1$ **to** $T$ **do**
        Choose action $a_t \sim \pi_\theta(s_t)$
        Observe $\boldsymbol{a_t}, \boldsymbol{r_t}, s_{t+1}$
        Store transition tuple $(s_t, a_t, r_t, s_{t+1})$ in $\mathcal{M}$ and joint transition tuple $(s_t, \boldsymbol{a_t}, \boldsymbol{r_t}, s_{t+1})$ in $\mathcal{H}$
    **end for**
    Update influence target function parameters $\phi$ with $\mathcal{H}$ every $k$ episodes
    Compute reciprocal rewards $r_{rc|i}^R(1), \ldots, r_{rc|i}^R(T)$ w.r.t. agent $i$ and sum with rewards in $\mathcal{M}$
    Compute advantage estimates $\hat{A}_1, \ldots, \hat{A}_T$
    **for** $K$ epochs **do**
        Optimize the surrogate PPO-clip objective w.r.t. $\theta_\pi$
    **end for**
    Reset $\mathcal{M} = \emptyset$
**end for**

---

influence over sequences of actions. We draw inspiration from the "debit" tally used by approximate Markov tit-for-tat (Lerer & Peysakhovich, 2018, amTFT), which tracks the cumulative advantage gained by an opponent compared to a fully cooperative baseline policy known *a priori*.

Extending this idea to our framework, we sum agent $i$'s influence $VI_{i|rc}^{\boldsymbol{\pi}}$ on the Reciprocator's expected return at each timestep rather than its own. Using influence in only one direction as motivation for reciprocation can lead to continuous punishment or rewarding without a way to settle the score. To mitigate this, we also accumulate the net influence $VI_{rc|i}^{\boldsymbol{\pi}}$ in the opposite direction and subtract it from the influence balance at every timestep as a way to "pay off" the balance, limiting the degree to which reciprocation is rewarded. Formally, we define the influence balance $B_{rc|i}(t)$ maintained by a Reciprocator $rc$ with another agent $i$ at time $t$ as

$$B_{rc|i}(t) = B_{rc|i}(t-1) + [VI_{i|rc}^{\boldsymbol{\pi}}(s_t, \boldsymbol{a}_t) - VI_{rc|i}^{\boldsymbol{\pi}}(s_t, \boldsymbol{a}_t)]. \tag{5}$$

The influence balance can be thought of as a score of net influence over time between agents, and can be used to motivate reciprocation in the correct direction, i.e., either positive reinforcement of net positive influence or positive punishment of net negative influence.

### 4.3  Intrinsic Reciprocal Reward

If agent $i$ takes a series of actions that improves the expected return of the Reciprocator over a baseline estimate, i.e., produces a net positive influence balance, then the Reciprocator should be motivated to reinforce this behavior by exerting a reciprocal positive influence on agent $i$'s expected return $R_i$, in order to encourage a higher likelihood of that behavior during policy updates. Similar logic applies for detrimental deviations, negative influence balance, and subsequent reciprocal punishment. We then define the intrinsic reciprocal reward $r_{rc|i}^R(t)$ received by $rc$ as

$$r_{rc|i}^R(t) = B_{rc|i}(t-1) \cdot VI_{rc|i}^{\boldsymbol{\pi}}(s_t, \boldsymbol{a}_t). \tag{6}$$

Taking the product of existing influence balance and current action's VI encourages the Reciprocator to take actions that reinforce agent $i$'s behavior in the correct direction by matching signs, and scales the reward by the magnitude of the outstanding influence balance and the reinforcing influence. The intrinsic reward is then added to the agent's extrinsic reward to form the total reward used in training.

### 4.4  Policy Optimization with Reciprocal Rewards

We use experience replay to periodically update target networks and iteratively update our counter-factual baseline estimates towards these target values (Mnih et al., 2015). This provides two key

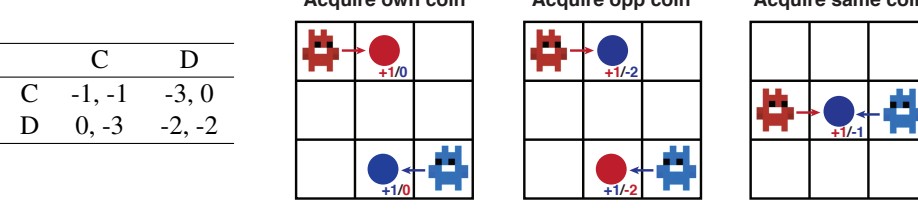

|   | C | D |
|---|---|---|
| C | -1, -1 | -3, 0 |
| D | 0, -3 | -2, -2 |

(a) Rewards for IPD.

(b) Rewards for Coins.

Figure 1: **(a)** The first number in each cell denotes the reward received by the agent taking the row action, and the second the reward received by the agent taking the column action, where C: cooperate (stay silent) and D: defect (confess). **(b)** Two agents (red and blue) are tasked with collecting randomly spawning coins. If an agent collects its own coin, it receives a reward of +1 (left). If an agent collects another's coin, then it receives a reward of +1 but the other agent receives a punishment of -2.

benefits: first, periodically updating these policy-dependent functions stabilizes training and allows us to approximately ignore their gradients, and therefore the gradient of the intrinsic reciprocal reward, with respect to the agents' policy parameters (Wang et al., 2020). With this assumption, we are able to use standard policy gradient methods to train our agents to jointly optimize the combined extrinsic and intrinsic rewards.

Second, drawing samples from multiple previous episodes to the train the counterfactual baseline target functions makes the Reciprocator less susceptible to exploitation. If the counterfactual estimators were updated concurrently with the policy using only the most recent on-policy data, then the Reciprocator's baseline would be immediately adjusted to its opponent's new policy after each episode. Because assessment of influence hinges on these counterfactual baselines, updating them too frequently would allow adversaries to easily manipulate these estimates of on-policy returns.

## 5 Experiments

We conduct experiments using two commonly used SSDs of varied complexity to demonstrate the shaping abilities of Reciprocators against other types of learning agents. For IPD, we consider memory-1 iterated games as in Foerster et al. (2018) and Lu et al. (2022), following the proof from Press & Dyson (2012) that longer-memory strategies provide no advantage over strategies conditioned on shorter memories. We use two methods to evaluate head-to-head performance in IPD: allowing agents to directly differentiate through the analytic, closed-form solution of the game as originally derived in Foerster et al. (2018), and more standard batched policy rollouts for a fixed episode length. We append "analytic" and "rollout" to the game names to denote the evaluation method used. For Coins, we augment the observation given to the critics and value influence estimators with the time remaining in the episode to prevent state aliasing and stabilize learning from experience replay (Pardo et al., 2022), but do not provide them as input to the policy networks.

### 5.1 Sequential Social Dilemmas

**Iterated Prisoners' Dilemma (IPD):** The iterated prisoners' dilemma (IPD) is a temporally extended version of the classical thought experiment, in which two prisoners are given a choice to either stay silent/cooperate (C) or confess/defect (D) with rewards given in Table 1a. The Pareto-optimal strategy is for both agents to cooperate by maintaining their silence, but the only Nash equilibrium (in the non-iterated, single-shot case) is mutual defection.

**Coins:** Coins is a temporally extended variant of the IPD introduced by Lerer & Peysakhovich (2018). In this game, two coins spawn randomly in a fixed-size grid, with each coin corresponding to one of the two players (designated by color matching). Each player moves around the grid and receives a reward of 1 for collecting any coin, and a punishment of $-1$ if the other agent collects their coin. If agents collect coins indiscriminately where $P(\text{collect own coin}) \approx 1/n$, the net expected reward is 0. We show example rewards for various scenarios in Figure 1b.

Table 1: IPD-Analytic round robin results

|        | RC     | NL     | LOLA   | M-MAML | MFOS   |
|--------|--------|--------|--------|--------|--------|
| RC     | -1.06  | -1.03  | -1.05  | -1.05  | -1.06  |
| NL     | -1.06  | -1.98  | -1.52  | -1.28  | -1.88  |
| LOLA   | -1.08  | -1.30  | -1.09  | -1.04  | -1.02  |
| M-MAML | -1.13  | -1.25  | -1.15  | -1.17  | -1.56  |
| MFOS   | -0.98  | -0.65  | -1.02  | -0.81  | -1.01  |

Each entry is the average reward per episode achieved by the row agent against the column agent. Standard error of the mean (SE) is less than 0.01 for all experiments and M-MAML results are averaged across the 10 initial policies provided by Lu et al. (2022).

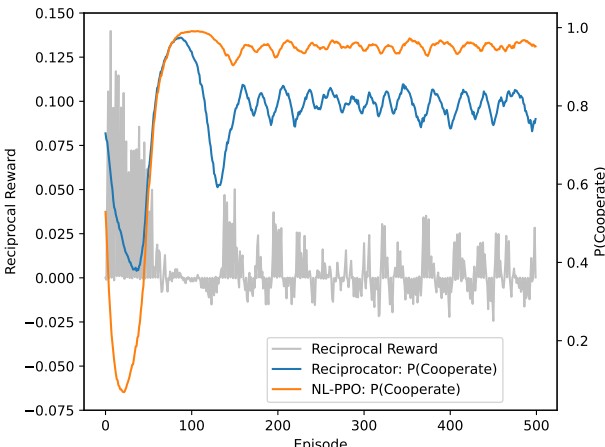

Figure 2: Representative run of a Reciprocator vs. an NL in IPD-Rollout. Average reciprocal reward per step (left axis) and probability of cooperation (right axis) over the course of an episode.

## 5.2 Baselines

We select the following baselines because they focus specifically on the problem of shaping other agents' policies during *simultaneous* learning and *without* modifications to the environment. In Coins, we exclude Meta-MAPG and LOLA-DICE as baselines following work by Yu et al. (2022) showing that neither method is able to achieve significant results, even with a simplified shared reward.

**Naïve Learner (NL):** As previously defined, NLs optimize their expected return with respect only to their own policy parameters $\theta$. In this work, we implement NLs using policy gradient methods (Sutton et al., 1999), which perform updates of the form $\theta_{t+1} = \theta_t + \alpha \nabla_\theta J(\pi_\theta)|_{\theta_t}$, where $\alpha$ is the learning rate and $\nabla_\theta J(\pi_\theta)|_{\theta_t}$ is the gradient of the objective function with respect to the policy parameters $\theta_t$ at step $t$.

**Learning with Opponent-Learning Awareness (LOLA):** LOLA uses either whitebox access to an opponent's gradients and Hessians or an explicit model of their policy parameters, assuming they are NLs, and differentiates through their learning step using the update rule

$$\theta_{t+1}^i = \theta_t^i + \alpha^i \nabla_{\theta_t^i} J^i \left( \theta_t^i, \theta_t^{-i} + \Delta\theta_t^{-i} \right)$$
$$\Delta\theta_t^{-i} = \alpha^{-i} \nabla_{\theta_t^i} J^{-i} \left( \theta_t^i, \theta_t^{-i} \right).$$

**Multiagent Model-Agnostic Meta-Learning (M-MAML):** M-MAML (Lu et al., 2022) learns initial parameters and then meta-learns over both its own and its opponent's policy updates, conceptually similar to Meta-Multiagent Policy Gradient (Kim et al., 2021, Meta-MAPG) but modified to differentiate directly through the analytic form of the return in matrix games.

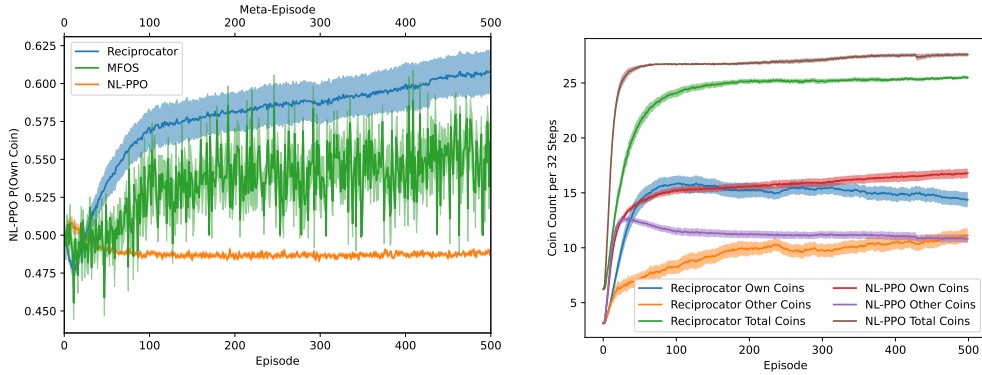

Figure 3: Shaping an NL in Coins. Proportion of own coins collected by NL during training when facing each opponent (left) and coin counts by type for Reciprocator vs. NL (right). Reciprocator and NL-PPO results are plotted on a scale of single episodes (bottom axis) whereas MFOS results are plotted on a scale of meta-episodes, where one meta-episode contains 16 episodes (top axis).

**Model-Free Opponent Shaping (MFOS):** As briefly discussed in the introduction, MFOS meta-learns over multiple episodes of policy updates in order to accomplish long-horizon opponent shaping. In Lu et al. (2022), MFOS is implemented with inner and outer policies, where the outer policy either directly outputs an inner policy to play in each episode or a conditioning vector which is element-wise multiplied with an inner policy vector, as done in IPD-Analytic and Coins, respectively.

### 5.3 Implementation Details

In IPD-Analytic, we differentiate directly through the analytic solution to the matrix game. For rollout-based experiments, we implement all policy gradient-based agents using actor-critic architectures trained with proximal policy optimization using a clipped surrogate objective (Schulman et al., 2017, PPO-Clip). Results including naïve learners trained with this algorithm are denoted by NL-PPO. Target networks to estimate the $Q$-values in Equation 1 are updated every $k$ episodes using uniformly sampled experience from a replay buffer (Lin, 1992). Additional hyperparameter values and network architecture details can be found in Appendix A.

## 6 Results

We evaluate the Reciprocators' performance against a variety of baselines in a round-robin tournament for IPD-Analytic. In Coins, we assess opponent-shaping performance against NLs and against another agent of the same type, which we refer to as symmetric Coins. Shaded regions indicate standard error of the mean (SE) over eight random seeds. We do not perform full round-robin experiments in non-analytic environments for the following reasons: the Reciprocators' complex and stochastic learning rule renders LOLA's assumptions invalid and the extended time to convergence due to reciprocal behavior makes collecting meta-training episodes for MFOS computationally prohibitive.

### 6.1 IPD

**IPD-Analytic**: In the analytic form of IPD, we show that Reciprocators are able to reach cooperative equilibria with all other baselines, resisting exploitation by higher-order methods such as LOLA and MFOS despite being only a first-order method using vanilla gradient descent [Table 1]. Although the Reciprocator is extorted by MFOS to small extent (-1.06 vs. -0.98, respectively), we emphasize that MFOS relies on extensive liberties such as the ability to observe thousands of parallel training runs against their opponents and roll out pre-trained meta-policies against newly initialized agents.

**IPD-Rollout**: Stochastic policy rollouts allow the Reciprocator to influence opponent returns differently for different action sequences and provide a stronger learning signal, motivating additional experiments using sampled rollouts. In this setting, we show that Reciprocator is able to consistently

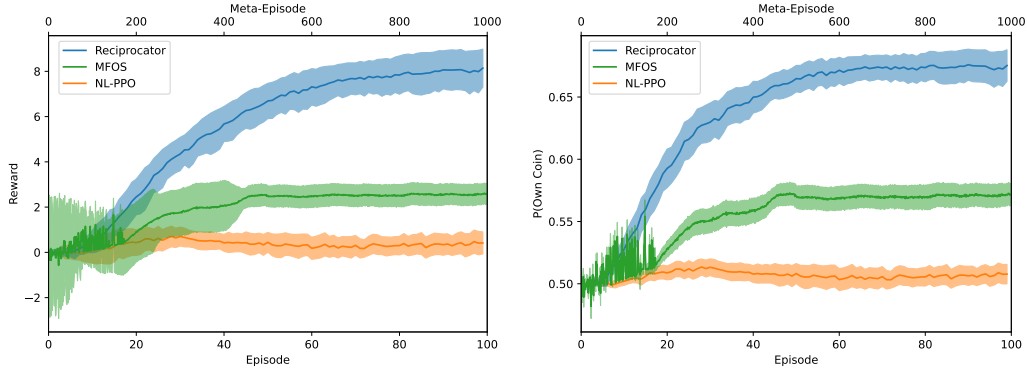

Figure 4: Head-to-head results in symmetric Coins (two agents of the same kind). Total (extrinsic) reward per episode (left), proportion of own coins collected (right). Again, Reciprocator and NL-PPO results are plotted on a scale of episodes and MFOS results are plotted on a scale of meta-episodes.

shape an NL over the course of a single learning trajectory with limited rollout samples and observe interesting oscillations in the Reciprocator's rate of cooperation. Figure 2 suggests that these oscillations are driven by the opposing intrinsic and extrinsic rewards, where the derivative of the rate of cooperation corresponds to the reciprocal reward value.

We provide the following explanation for this phenomenon: although the Reciprocator initially learns a cooperative tit-for-tat strategy guided by a strong reciprocal reward, as the NL's policy becomes deterministic and $a^i$ becomes predictable, the $VI^{\pi}_{i|rc}$ component of the influence balance decreases to 0. As the extrinsic reward begins to dominate, the Reciprocator essentially reverts into an NL, leading to exploitative defection. However, exploitation of the NL agent $i$ causes the $VI^{\pi}_{rc|i}$ term, and subsequently the reciprocal reward, to become negative. Combined with an increase in the NL's frequency of defection, which leads to an increase in $VI^{\pi}_{i|rc}$, the intrinsic reward produces a reversal back to cooperative behavior that is reinforced by positive reciprocal rewards, and the cycle repeats.

## 6.2 Coins

In this temporally extended social dilemma, we see that the Reciprocator is able to shape NL-PPO into picking up more of its own coins at rates significantly higher than MFOS. This allows both agents to achieve a positive reward, while needing only a fraction of the samples to converge to mutually beneficial behavior (each meta-episode consists of 16 sequential episodes) [Figure 3]. We show that this change is driven by changes in coin preference rather than in total collection, with both agents collecting at near-optimal pace. This is in contrast to MFOS, which does not shape the NL towards cooperation, but rather uses its pretraining advantage to suppress opponent learning altogether (Khan et al., 2023) by collecting coins faster than the NL can reach them [Appendix B.1].

When two Reciprocators are pitted against each other, we see that they quickly learn a cooperative strategy of collecting their own coins [Figure 4], resulting in an average reward of $\sim 8$ per 32 steps *without needing* self-play. This significantly outperforms MFOS and the reported performance of LOLA with opponent modeling (LOLA-OM), which achieves an average reward of only $\sim 2$ per 32 steps according to Foerster et al. (2018). While it can be argued that having intrinsic rewards for both opponents in the symmetric setting effectively alters the reward structure of the social dilemma into a cooperative game, we emphasize that the reciprocal reward encourages opponent-shaping behavior that can take the form of a cooperative strategy, rather than explicitly incentivizing cooperation for cooperation's sake.

Together, these results demonstrate that Reciprocators are able to robustly shape the behavior of other agents towards prosocial equilibria during simultaneous learning, achieving state-of-the-art results with fewer assumptions and limitations than existing methods. Apart from opponent-shaping properties, we also show that the intrinsic reciprocal reward discourages Reciprocators from exploiting others, showing promise for the development of a more cooperative multi-agent learning framework.

# 7 Limitations and Future Directions

Our implementation of Reciprocators using naïve RL algorithms remains limited in that it seeks to maximize its compound return within single episodes rather than across multiple episodes. Using handpicked weights to balance intrinsic and extrinsic rewards opens the door for a suboptimal tradeoff between long-term opponent shaping and short-term return maximization. This is partially mitigated by the counterfactual target baseline updates, which allow the Reciprocator to lower the magnitude of the reciprocal reward in response to opponent policies that remain stationary over multiple episodes [Figure 2]. Future work will focus on methods to evaluate reciprocation efficacy across multiple episodes and dynamically tune the balance between reciprocal and extrinsic rewards.

In terms of evaluation, Khan et al. (2023) found that Coins does not require history to enable opponent shaping, since the current state is often indicative of past actions. Due to computational limitations, we were unable to assess our method's performance on the suite of Spatio-Temporal Representations of Matrix Games (STORM) designed to test shaping over longer time horizons. We do note that Reciprocators capture both types of memory necessary to achieve shaping as identified by Khan et al. (2023): the counterfactual baselines trained on replay buffers serve as a way to capture inter-episode context, and the recurrent policies and influence balance capture intra-episode history. Therefore, our method can in theory generalize to STORM environments, although we leave this to future work.

# 8 Conclusion and Broader Impacts

Emerging interest in cooperative AI has been led by approaches that endow agents with higher-order shaping capabilities. Although these agents exhibit cooperative behaviors when pitted against equivalent opponents, they readily manipulate and exploit agents with simpler learning rules or lower computational capabilities. This is a fundamentally undesirable outcome in partially adversarial interactions between learning agents, with the potential to exacerbate existing computational resource gaps beyond out-of-the-box pretrained performance.

We presented Reciprocators, agents which seek to influence the behavior of other, simultaneously learning agents towards mutually beneficial outcomes. We showed that Reciprocators can both learn prosocial behaviors and induce them from other agents in a variety of sequential social dilemmas, while remaining resistant to exploitation by higher-order agents. To the best of our knowledge, Reciprocators represent the first class of reinforcement learning algorithms to achieve cooperation during simultaneous learning between two independent agents *without* needing meta-learning methods, knowledge of other agents' learning algorithms, or pretraining routines such as self-play or tracing procedures to control opponent selection. We believe that these results show a promising avenue forward for inducing cooperative outcomes from a diverse array of learning opponents.

## Acknowledgements

This work was supported by the following awards to JCK: National Institutes of Health DP2NS122037 and NSF CAREER 1943467.

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

# A  Experimental Details

All experiments were run on Nvidia 3070 GPUs with 8 GB of VRAM. Results were averaged over eight random seeds, with a batch size of 8192 for IPD-Analytic and 2048 for IPD-Rollout and Coins. For all experiments involving MFOS agents, we adapted the original code from Lu et al. (2022), leaving all architectural choices and hyperparameters the same.

The architecture for both the actor and the critic consists of a state encoder (two convolutional layers followed by a linear layer, with ReLU activations between layers), followed by hidden linear layers as detailed in the table below and a final output linear layer. The actor has a softmax activation to output a policy over the action space, whereas the critic simply outputs a scalar value estimate.

Table 2: General PPO parameters.

| Hyperparameter | IPD-Rollout | Coins |
|---|---|---|
| Number of Convolutional Layers | - | 2 |
| Convolutional Kernel Size | - | 3 |
| Number of Linear Layers | 2 | 1 |
| Size of Linear Layers | 2 | 16 |
| Number of GRUs | - | 1 |
| Size of GRUs | - | 16 |
| Episode Length | 32 | 32 |
| Adam Learning Rate | 0.005 | 0.005 |
| PPO Epochs Per Episode $K$ | 10 | 40 |
| PPO-Clip $\epsilon$ | 0.1 | 0.15 |
| Discount Factor $\lambda$ | 0.96 | 0.96 |
| Entropy Coefficient | 0.02 | 0.01 |

With the exception of a linear layer size of 32 (instead of 16) for Coins, the network architecture and parameters to estimate the various target functions to compute the $VI$ are identical to those described for the PPO components in Table 2. For IPD-Analytic, target estimates for opponent policies in IPD were computed as the frequencies of observed choices for each of the five possible states (start, CC, CD, DC, DD).

For Coins, we implement the decomposition of $VI$ given in Equation 2, separately predicting the immediate reward and the transition probabilities for each possible next state. We collect one batch of episodes at the beginning of each experiment to initialize the influence estimators.

Table 3: Reciprocator-specific parameters.

| Hyperparameter | IPD-Analytic | IPD-Rollout | Coins |
|---|---|---|---|
| Replay Buffer Size (in episodes) | 5 | 1 | 4 |
| Target Training Batch Size | - | - | 4096 |
| Batches per Epoch | - | - | 64 |
| Target Function Update Period | 10 | 3 | 1 |
| Target Function Epochs Per Episode | - | - | 20 |
| Adam Learning Rate | - | - | 0.01 |
| Reciprocal Reward Weight | 5.0 | 5.0 | 1.0 |

Replay buffer sizes are in units of episodes, where each episode consists of a batch of 32 steps. A replay buffer of size 4 for Coins corresponds to 32 steps $\times$ 4 episodes $\times$ 2048 batch size = 262,144 steps of experience.

# B  Additional Results

## B.1  Coin Counts vs. NL-PPO

We display the total coin counts of an NL when faced against each other type of agent in order to show that changes in P(Own Coins) when faced by a Reciprocator are driven by changes in coin

color preference rather than changes in the total number of coins collected. In contrast, we see that the NL collects far fewer coins when faced with an MFOS agent, providing support for Khan et al. (2023)'s claim that MFOS suppresses rather than shapes learning. Note that this figure corresponds to the same experimental data as Figure 3.

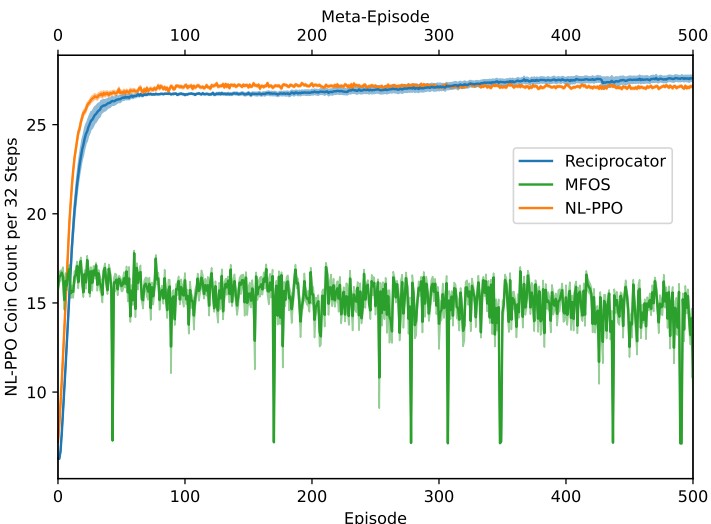

Figure 5: Total number of coins per 32 steps collected by NL-PPO (right) vs. each baseline in Coins.

