# OpenReview forum: "Reciprocal Reward Influence Encourages Cooperation From Self-Interested Agents"
_NeurIPS.cc/2024/Conference — NeurIPS 2024 poster_

### Official Review · Reviewer_os8m · 2024-07-07

**Soundness:** 4
**Presentation:** 4
**Contribution:** 3
**Rating:** 6
**Confidence:** 3

**Summary:**

The paper proposes a method for including the influence of other agent’s into an agent’s own reward function, which is shown to promote cooperation in SSDs like IPD and the temporal version Coins. It also highlights that the method only requires first order RL algorithms, does not require access to privileged information, and does not require meta-learning.

**Strengths:**

- The paper is well written.
- The papers has clear comparisons with state-of-the-art methods. Despite lower scores in IPD against MFOS, the authors are clear to state that improvements include not only higher scores, but also in a reduction of steps required to reach cooperation, reduced complexity in opponent shaping methods, and the ability to cooperate without access to privileged information. First order RL algorithms and standard roll out processes make reproducibility and adoption of the method easier than higher order methods requiring privileged information or meta-learning.

**Weaknesses:**

- Not all results are shown clearly for IPD-Rollout or Coins. I suggest a result matrix similar to IPD-Analytics or a clear explanation as to why it is not included, even if only in the Appendix.
- Nit: Add titles to Section 5 Figure 2 and 3.
- Nit: Section 5.2 Figure 2 description: percentage -> probability.

**Questions:**

- LOLA vs. MFOS in Section 5.2 Table 1 does not match the original MFOS paper results. In MFOS Section 6.1, Table 4, LOLA vs. MFOS resulting in -2.09 for LOLA, while your section claims -1.02. Your result implies that LOLA has learned to cooperate with MFOS, when the MFOS paper shows that MFOS should learn a dominating strategy against LOLA. Where is the discrepancy coming from here?
- Section 6 Figure 4 — Does each agent in this symmetric head-to-head receive the same reward and have the same probability of own coins?
- Is there an experiment for Reciprocator vs. MFOS in Coin Game?
- L 469: It’s unclear why the mean intrinsic reward begins as a negative number and converges to 0. I would have expected the intrinsic reward to decrease from a positive amount down to 0.
- L 293: Is it possible that implementation details created a difference?
- Will the codebase be made available upon acceptance? Concrete examples for your paper’s method will help future researchers implement baselines in accordance with your paper.

**Limitations:**

Yes.

---

> ### Author Rebuttal · Authors · 2024-08-06
>
> Weaknesses
>
> W1
>
> -   One key issue with LOLA and IPD-Analytic is that the original LOLA implementation assumed that the opponent takes vanilla gradient steps directly through the closed-form analytic solution to the IPD return. LOLA then directly computes the gradient of the opponent's return w.r.t. the opponent's parameters to use for its update. This approach does not work in our case, where the Reciprocator's policy update takes into account the intrinsic reward (which is recurrently computed at each step), precluding a closed-form solution to IPD. Since the rollout environment introduces stochasticity rather than analytic solutions, we found that stochastic optimizers such as Adam provided much better results, at the expense of a fixed update rule. Because of this, we did not think it would be a fair comparison/equivalent evaluation to create a new,  "hamstrung" formulation of LOLA that is differentiating through an incorrect gradient update and present it as a baseline. We will make these choices clear in the Appendix.
>
> -   Unfortunately, running MFOS against the Reciprocator is computationally intractable in Coins (we elaborate on this in our response to Q3), and LOLA has failed to show significant results in Coins, even with a simplified shared reward [1]. We mention the latter in the beginning of Section 6, but will be sure to add in our rationale for only doing symmetric and not all head-to-head comparisons for Coins and provide more detailed statistics for the remaining experiments in the Appendix.
>
> -   We do want to emphasize that, while these factors reduced the scope of our evaluation, they also point to the clear limitations on existing opponent shaping methods and the relative generality of our simple intrinsic reward.
>
> W2
>
> -   We will add appropriate titles to these figures, thank you!
>
> W3
>
> -   We have made this fix, thank you!
>
> Questions
>
> Q1
>
> -   We are not sure of the discrepancy between these results, as we reproduced the IPD-Analytic results by directly running their published code on Github with no changes, averaged over 8 runs. However, we will investigate further to see if there might be differences in averaging or random seed that may explain this large gap.
>
> Q2
>
> -   Agents in symmetric head-to-head games do receive similar rewards and cooperate with similar probabilities. We provide a representative run in Appendix Figure B.1 (incorrectly linked as A5 in the text) and attempt to communicate this in Figure 4 by providing the standard error of the mean computed over both agents in symmetric games. If the two agents achieved significantly inequitable outcomes, then there would be much higher variance in these figures, i.e., the shaded areas would be much wider. We have clarified this in the figure legend.
>
> Q3
>
> -   We did not provide a result for Reciprocator vs. MFOS since it is computationally infeasible to do so: as a meta-learning algorithm, MFOS treats full training runs to convergence as *single episodes*, requiring not only exponential sample complexity [2] but also an order of magnitude more gradient updates for each agent (num_meta_episodes x num_episodes x num_epochs). Reciprocators take significantly longer to converge than NLs due to their intrinsic opponent shaping objective and additional network updates. To provide a fair comparison, we would need to train at least 16 runs to convergence *sequentially*, which would require training times on the order of hundreds of hours for just 8 replications, assuming no hyperparameter tuning. We did perform one Reciprocator vs. MFOS experiment and found that MFOS failed to shape the Reciprocator or learn in any meaningful way, but due to the lack of replication power we did not include it in our paper - however, we would be happy to run these experiments for a camera-ready copy.
>
> Q4
>
> -   We attribute this to the conflicting motivations of the extrinsic and intrinsic reward. The extrinsic reward pushes the agent to naively collect all coins regardless of color, whereas the intrinsic reward encourages the agent to selectively collect coins depending on the influence balance: collecting the opponent's coin when punishment is desired, and avoiding it when it is not. In the case where the extrinsic reward initially dominates, the mean intrinsic reward should be negative since the the agent may be collecting coins at suboptimal times w.r.t. reciprocation. For example, if the influence balance is positive, the Reciprocator should avoid collecting the other agent's coins and therefore punishing it -- however, the extrinsic reward encourages it to take any coin as fast as possible, which would lead to a negative reward if the closest coin happens to be that of the opponent's.
>
> Q5
>
> -   We constructed our agent based on the MFOS codebase and with hyperparameters as similar to the LOLA agent as possible, matching the reported architecture (including all layer sizes and activations), learning rate, discount factor, and environment implementation. We use a smaller batch size (2048 vs. 4000) due to memory constraints. They used a vanilla actor-critic method, which does not have the clipped gradient of PPO but is otherwise similar. We also point to subsequent work by [1], which found that LOLA-DiCE (LOLA with an improved differentiable Monte Carlo estimator) was unable to model opponents quickly enough and thus failed to achieve significant opponent shaping in Coins against a naive learner.
>
> Q6
>
> -   The codebase is already available as a zipped attachment in the supplementary materials, and will also be linked as a Github repository in the paper after the blinded review.
>
> References
>
> [1] Yu, X., Jiang, J., Zhang, W., Jiang, H. & Lu, Z. Model-Based Opponent Modeling. (2022).
>
> [2] Fung, K., Zhang, Q., Lu, C., Willi, T. & Foerster, J. N. Analyzing the Sample Complexity of Model-Free Opponent Shaping. (2023).

---

> > ### Comment · Reviewer_os8m · 2024-08-07
> >
> > Thank you for your detailed response to my concerns. I have read your rebuttal and will provide further comments soon.

---

> > > ### Comment · Reviewer_os8m · 2024-08-11
> > >
> > > The authors have clarified my questions and I would like to maintain my score.

---

### Official Review · Reviewer_PQwp · 2024-07-12

**Soundness:** 3
**Presentation:** 4
**Contribution:** 3
**Rating:** 6
**Confidence:** 5

**Summary:**

The authors develop a new intrinsic reward that tracks the balance of influence compared to a counterfactual baseline and provides positive/negative rewards to incentivize naive learners to cooperate. They test this algorithm in an iterated prisoner dilemma and a simplified coin game, showing positive results compared to some strong baselines on opponent shaping.

**Strengths:**

This is a paper addresses an important topic in multi-agent reinforcement learning that will be of active interest to the NeurIPS community. The manuscript is well written and the results are well situated within the current literature. The algorithm is benchmarked against strong modern baselines and the results are discussed appropriately and clearly.

**Weaknesses:**

- There are some missing implementation details (number of seeds, averaging methods).
- I’d have liked to see more extensive evaluation on the grid environments where the RL results are more important.
- The evaluation could be more robust and compare in more sophisticated games. Even the coin game is highly simplified compared to the original versions.
- While the algorithm is novel in the RL literature this is highly reminiscent of the following work on commutative recipcority which develops analytical results for a similar approach. Ideally the methods here could be compared to the algorithm and analyses proposed in this work: Li, J., Zhao, X., Li, B., Rossetti, C. S., Hilbe, C., & Xia, H. (2022). Evolution of cooperation through cumulative reciprocity. Nature Computational Science, 2(10), 677-686.

**Questions:**

- While MarkovTFT is discussed — how is this approach significantly different? Given that the algorithm learns a TFT like algorithm — when would this approach be preferred and how does it advance the literature? The authors should also discuss a more probabilistic approach to TFT published in this work: Kleiman-Weiner, M., Ho, M. K., Austerweil, J. L., Littman, M. L., & Tenenbaum, J. B. (2016). Coordinate to cooperate or compete: abstract goals and joint intentions in social interaction. In CogSci.
- The opponent shaping work has the potential to shape towards a wide variety of goals and aims — this work seems more focused on just cooperation. Is that right? If so — this should be discussed in more detail as a limitation.
- Unless the algorithm correctly models what it could be doing counterfactually — it won’t know how to accurately update the balance of influence. Will this always be learned rapidly? Could an adversary bias its learning away from forming an accurate model?

**Limitations:**

Yes

---

> ### Author Rebuttal · Authors · 2024-08-06
>
> Weaknesses
>
> W1
>
> - We have added these details in the Appendix: each experiment was run with 8 random seeds, and results were averaged and plotted with SEM.
>
> W2-3
>
> - To the best of our knowledge, previous works in the opponent shaping literature have conducted evaluations only on simple iterated matrix games and this simplified version of Coins, so we focused on these settings in order to provide baseline comparisons (MFOS would be especially computationally prohibitive in more complex environments--see Lu et al. 2022). However, we concur that these evaluations are simple compared to the broader RL literature and plan to demonstrate the merits of our approach on more complex games in future work.
>
> W4
>
> - Our work can be considered a general implementation of a "cumulative reciprocator" (CURE). The CURE strategy depends on discrete, countable instances of cooperation and defection, which are used to compute the defection difference statistic d(k). While this works in simple matrix games such as IPD, it is more difficult to directly classify individual actions as cooperative or defective in temporally-extended games such as Coins, inspiring our use of the modified VI metric to measure continuous changes to expected return. Our influence balance is also similar to the defection difference statistic, although we subtract VI's at every timestep to directly track this statistic rather than maintaining running tallies.
>
> -   We find their analyses on the role of the tolerance level in increasing cooperation stability vs. noisy decisions particularly interesting as potential future directions for our work, e.g. adding an influence balance threshold to the reciprocal reward.
>
> -   We will include this discussion in our related works. As for the payoff analyses, we note that CURE is a fixed strategy that does not have to consider learning dynamics. However, our method seeks to influence the learning of other, simultaneously learning agents. Therefore, any solution must consider learning algorithms and associated parameters, making closed-form solutions far more difficult (if even possible) to derive.
>
> Questions
>
> Q1
>
> - Our approach can be considered a more general version of amTFT, which seeks to achieve cooperation in SSDs by combining two pretrained strategies: one cooperative and one defective (note that defining such pure strategies is rarely possible in more complex social dilemmas). amTFT monitors the partner's gains from deviating from a cooperative baseline (debit) and switches to defection as punishment once the debit crosses a threshold. While our reward draws inspiration from amTFT, amTFT relies on pretrained policies and handpicked parameters (the defection duration and debit threshold) and focuses on performance against fixed opponents or as teachers for naive learners. It does not tackle the problem of emergent cooperation among simultaneously learning agents -- after all, it is not a learning agent itself but a composition of frozen components. On the other hand, our reciprocal reward encourages agents to learn when and how to cooperate or defect in an end-to-end policy. Notably this is learned in an environment with other, simultaneously learning agents, presenting a significantly more difficult, nonstationary learning problem.
>
> - Kleiman-Weiner et al. also learn a hierarchical policy consisting of cooperative and defective "modes." A high-level planner infers the high-level intention (I) of an opponent and responds in kind (using a TFT strategy or learned high-level policy) by selecting between two sub-strategies. Notably, they use a low-level dynamics model to both plan sub-strategies and infer opponent intentions using value iteration, which requires modeling of both of the opponent's sub-policies and inherently assumes that they are following either a pure cooperative or defective strategy at any given time. On the other hand, our algorithm makes no such assumptions on the "purity" of strategies, the intent behind them, or their homogeneity throughout an episode, instead operating solely on the actual effects of the opponent's behavior on the Reciprocator's expected returns to dynamically shift between cooperative and defective behavior.
>
> Q2
>
> - Although the Reciprocator does focus on cooperation, it can be extended to other shaping goals by modifying the $VI_{i|rc}$ formulation to compute the influence of agent i's action on some other measure. The key mechanism is that the intrinsic reward motivates the agent to alter the returns of the other agent in a specified direction following an action - matching the direction of alteration to other criteria against which opponent actions can be evaluated would permit more general shaping goals.
>
> - We will expand on this limitation as well as the potential for alternative shaping goals in the Future Directions section.
>
> Q3
>
> - This is a great question! Another adversary with a meta-policy operating across episodes could launch an adversarial attack by manipulating the baseline into producing overly pessimistic estimates, e.g., by maintaining a highly defective policy for multiple timesteps to "lower expectations" of on-policy returns, s.t. even objectively harmful future actions may be viewed as having a positive influence and induce unwarranted reciprocation. This method of exploitatively lowering expectations is a common manipulation tactic in real-life, e.g., "weaponized" incompetence.
>
> - Performing updates of the counterfactual baseline too rapidly can make the Reciprocator more susceptible to such attacks. By periodically updating the baseline with experience across multiple episodes, we allow the Reciprocator to maintain an inter-episodic form of memory that prevents opponents from rapidly manipulating baseline expectations within single episodes. As a future direction, we propose to preferentially select episodes with higher return using prioritized replay in order to produce optimistic counterfactual baselines.

---

> > ### Comment · Reviewer_PQwp · 2024-08-11
> >
> > Thank you for the detailed response. I will discuss these responses with the other reviewers. I have no further questions at this time.

---

### Official Review · Reviewer_SrMN · 2024-07-12

**Soundness:** 2
**Presentation:** 2
**Contribution:** 1
**Rating:** 3
**Confidence:** 4

**Summary:**

The paper discusses an analysis of the effects of a combination of methods (1-step influence, debt mechanism and intrinsic reciprocal reward) on the emergence of cooperation in a society of artificial agents. The authors evaluate their approach for the case of n=2 using an IPD and the Coin game.

**Strengths:**

- The reviewer personally believes that the study of cooperation in multi-agent systems is both interesting and timely. Despite the significant amount of work already conducted in this area, the reviewer welcomes further research and contributions

**Weaknesses:**

- The contribution of each mechanism (1-step influence, debt mechanism, and intrinsic reciprocal reward) is not discussed and evaluated in the paper. In fact, the author essentially combines them without clearly showing evidence that all three are actually needed. There is also a key question about the actual improvement that is possible to obtain by introducing them (and with which parameters). The mechanisms themselves are not new. Value influence comes from (Wang et al., 2020) and the idea of debt comes from (Lerer & Peysakhovich, 2018). That is perfectly fine, but I think it is important to understand if and how they are needed for observing the emergence of cooperation.

- In general, the reviewer would have welcome some discussions (in terms of intuitions) about the design choices that are at the basis of this paper. In fact, it is not completely clear *why* the proposed mechanisms should work (and also *how* in a sense).

- The mechanism presented by the authors appears to be somewhat related to the idea of social learning presented in [1]. However, this work is not discussed by the authors, even though it seems closely related. I believe a discussion about the relationship between that class of approaches and the one presented in this paper would be beneficial.

- It would be interesting to see a comparison with other mechanisms that foster the emergence of cooperation, particularly in the evaluation of the proposed method. Instead, the authors present comparisons with algorithms (like LOLA), which seem unrelated to the problem at hand. In fact, we know beforehand that these mechanisms do not lead to the emergence of cooperation.

- The theory presented in the paper essentially refer to a multi-agent scenario. However, the authors present their results only for the case of $n=2$. This case is not really informative (also comparing with existing works in the literature cited by the authors). The emergence of cooperation with $n=2$ is a phenomenon that has characteristics quite different compared to the case of $n>2$ in my opinion.


References

[1] Jaques N, Lazaridou A, Hughes E, Gulcehre C, Ortega P, Strouse DJ, Leibo JZ, De Freitas N. Social influence as intrinsic motivation for multi-agent deep reinforcement learning. In International Conference on Machine Learning 2019 May 24 (pp. 3040-3049). PMLR.

**Questions:**

- Do you have any results in terms of “ablation” considering the presence or not of the various mechanisms presented in the paper?

- How does your work relate to the idea of social learning and the use of sort of social influence as intrinsic reward?

- How does your work compare with Hughes et al. 2017? In fact it seems to me that the underlying mechanism might be similar to inequality aversion in a sense.

- In Equation (1), the reviewer is not sure if there should be a sum over $a_i$: that appears to be an error. Could you please clarify the mathematical formulation?

- Also in Equation (2), the sum over $a_i$ does not appear to be correct. Could you please discuss this formula as well?

- In Section 4.4. you listed two “assumptions”. The reviewer wonders if it would have been preferable to validate them. Do you have results supporting these assumptions?

- Why did you select LOLA, M-MAML, etc. as baselines? They do not appear to be relevant in a study about the emergence of cooperation. In fact, it seems to me that we know a priori that they do not lead to cooperation. It would have been more helpful to show comparison with other papers (like those that are cited in the related work) that actually lead to the emergence of cooperation.

- Do you have results for the case of n>2? In fact, the theory is presented for n>2, but the evaluation is carried out for the case of n=2, which is very limiting and not sufficiently informative in my opinion.

**Limitations:**

There is not a real discussion of the limitations of the work in terms of generalization to the case of the proposed approach to the case of $n>2$ (in terms of evaluation).

---

> ### Author Rebuttal · Authors · 2024-08-06
>
> Weaknesses
>
> W1 - 2
>
> - We apologize for our lack of clarity and hope the following provides intuition: reciprocating strategies (such as TFT) are known to induce cooperation [1]. Reciprocation requires the ability to distinguish cooperation from defection, which is difficult to define *a priori* in SSDs. This inspired our adaptation of VoI [2], which allows us to assess the "cooperativity" of an action without predefined notions of cooperation/defection.
>
> - However, agents often have limited agency in SSDs, so they may not be able to immediately reciprocate. Keeping a running tally of influence over time motivates the agent to reciprocate when it has the opportunity to do so. To keep this tally, we constructed the influence balance inspired by debit [3].
>
> - The intrinsic reciprocal reward is a mechanism to implement a tit-for-tat (TFT)-style strategy using the aforementioned components, where the product of the influence balance and the VI of the current action is positive for reciprocating actions.
>
> - If this is a helpful clarification, we will be glad to revise our manuscript to include these intuitions.
>
> W3 - W5
>
> - Addressed in question responses.
>
> Questions
>
> Q1
>
> - Because our three mechanisms are combined nonlinearly into a single reward term, it does not make sense to ablate individual parts. For example, if we ablate the influence balance, we cannot determine the correct direction of reciprocation. Additionally, since these components have not been used in the context of opponent shaping, there are no suitable "replacements" that can serve as an ablation baseline.
>
> Q2
>
> - While both approaches can be considered as forms of social influence, Jaques et al. seek to influence the actions of other agents via a mutual information objective, without considering how the behavior is modified. This undirected measure of social influence has unclear utility, and may even be meaningless in cases where different actions lead to equivalent rewards or transitions.
>
> - On the other hand, our intrinsic reward explicitly considers how opponent actions affect the Reciprocator's expected returns, and vice versa. It encourages the agent to influence the opponent's returns in a specific direction for opponent-shaping purposes.
>
> - We will include a discussion of these similarities and distinctions in our Related Works.
>
> Q3
>
> - Inequity aversion differs from our method in that it adds a prosocial objective by penalizing agents who under- or over-perform relative to others. Conversely, our reciprocal reward is not inherently prosocial, but seeks to encourage other agents to take actions which benefit the Reciprocator as an opponent-shaping mechanism.
>
> - A key result of this difference: Hughes et al. note that their "guilty" agents are easily exploited if the population is not composed **only** of guilty agents (see L60 - 65). However, the opponent-shaping properties of Reciprocators allow them to induce cooperative behavior from purely self-interested agents while conferring a robust resistance to exploitation against other shaping agents, e.g., MFOS.
>
> Q4
>
> - We clarify that Equation (1) is correct. Please note that we define the 2nd term, $Q(s, a^{-i})$, as an expectation/sum over $a^i$'s in Equation (3), which we believe addresses the confusion.
>
> Q5
>
> - Similar to the definition of the counterfactual Q-function in Equation (3), the counterfactual $r(s, a^{-i})$ should be an expectation over $a^i$ and the summation over states should also be over $a^i$. We have defined $r(s, a^{-i})$ and fixed the summation term in our revision.
>
> Q6
>
> - The first assumption is the same justification used for the original VoI from Equation 13 in [2], where the replay buffer and periodic update of the counterfactual target functions makes the gradients of these counterfactual estimators (and therefore the intrinsic reward) approximately constant w.r.t. to policy parameters. Because of its validation in previous work, we did not think it necessary to validate them here. We will update the manuscript to clarify this.
>
> - As for the second claim, we found that immediate updates using only the most recent on-policy experience from the last episode produced training instabilities resulting in lopsided exploitation - we will add results to the Appendix.
>
> Q7
>
> - We respectfully disagree with this point - one of the key results of LOLA is that they learn to cooperate: "LOLA agents **learn to cooperate**, as agents pick up coins of their own colour with high probability while naive learners pick up coins indiscriminately" [1]. Similarly, MFOS notes that it is "the first learning algorithm **to achieve cooperation** in self-play without using higher-order derivatives or inconsistent models" [4].
>
> - Because the goal of the reciprocal reward is to shape opponent behavior towards self-benefiting actions, we see our work as an opponent-shaping method similar to LOLA and MFOS which can lead to emergent mutual cooperation, and distinct from MARL algorithms which have the sole purpose of cooperation such as Hughes et al. (2018).
>
> Q8
> - While the $n > 2$ setting introduces increased complexity, cooperation in the $n = 2$ setting with a purely self-interested opponent in an *unmodified environment* remains nontrivial. Additionally, and to the best of our knowledge, there are no works in the literature that attempt to shape multiple opponents simultaneously. Therefore, we chose these settings for compatibility with modern opponent shaping baselines. However, we concur that these evaluations are simple compared to the broader RL literature and will demonstrate the merits of our approach on more complex games in future work.
>
> - We also reiterate that methods which have induced cooperation among $n > 2$ agents, such as the "guilty" agents of Hughes et al. (2018), require homogeneous prosocial populations and are easily exploited otherwise, whereas shaping methods such as [5] modify the environment to allow direct influence over others' rewards [5].

---

> > ### Comment · Reviewer_SrMN · 2024-08-10
> >
> > I would like to thank the authors for their rebuttal. However, the reviewer still has major concerns about this paper, especially with respect to the contribution of each specific mechanism and their inter-play. Essentially, it is unclear if all these mechanisms (and inherent complexity) are necessary.
> >
> > With respect to the baselines, it seems to me that the type of “cooperation” in LOLA/MFOS is different from that defined by the authors. The authors of LOLA use “cooperation” as a term, but it is a different phenomenon from the emergence of cooperation of interest for the authors.
> >
> > With respect to the notation, I see what the authors mean, but I would suggest trying to improve/correct the notation in terms since those terms are not essentially defined in the formula, in my opinion.
> >
> > For these reasons, I will maintain my assessment of this work.

---

> > > ### Comment · Reviewer_SrMN · 2024-08-10
> > >
> > > It seems that the authors also used two boxes for the answer - not sure if it is over the limit, I cannot check from the interface (for a general matter of fairness).

---

> ### Author Response · Authors · 2024-08-07
> **References**
>
> [1] Foerster, J. N. et al. Learning with Opponent-Learning Awareness. in Proceedings of the 17th International Conference on Autonomous Agents and Multiagent Systems (2018). doi:10.48550/arXiv.1709.04326.
>
> [2] Wang, T., Wang, J., Wu, Y. & Zhang, C. Influence-Based Multi-Agent Exploration. in Eighth International Conference on Learning Representations (2020).
>
> [3] Lerer, A. & Peysakhovich, A. Maintaining cooperation in complex social dilemmas using deep reinforcement learning. Preprint at https://doi.org/10.48550/arXiv.1707.01068 (2018).
>
> [4] Lu, C., Willi, T., Witt, C. A. S. D. & Foerster, J. Model-Free Opponent Shaping. in Proceedings of the 39th International Conference on Machine Learning 14398–14411 (PMLR, 2022).
>
> [5] Yang, J. et al. Learning to Incentivize Other Learning Agents. in Advances in Neural Information Processing Systems vol. 33 15208–15219 (Curran Associates, Inc., 2020).

---

> ### Author Response · Authors · 2024-08-10
>
> As stated in the clarifying email sent by the conference PCs, “**Comments to paper and reviews will be fine. Comments can be seen in time. Please set the readers correctly when you post them. Reviewers are not required to take comments into consideration**.” However, we note that our follow-up comment was simply a list of references to the main rebuttal, making this a relatively minor point.

---

> ### Author Response · Authors · 2024-08-10
>
> > However, the reviewer still has major concerns about this paper, especially with respect to the contribution of each specific mechanism and their inter-play. Essentially, it is unclear if all these mechanisms (and inherent complexity) are necessary.
>
> -   Once again, we apologize for our lack of clarity and will use two motivating examples from the literature to further establish the basis of not just one specific mechanism, but all of them combined together to produce a single TFT-like strategy.
>
> -   We reference amTFT, which seeks to achieve cooperation in SSDs by combining two pretrained strategies: one cooperative and one defective. amTFT monitors the partner's gains (debit) from deviating from a cooperative baseline and switches to defection for a fixed amount of time $k$ as punishment once the debit crosses a threshold, s.t. any gains from defection are fully neutralized. Our work takes a similar TFT-like approach, but tackles two key problems: 1) it is rarely possible to classify actions as purely cooperative or defective in temporally-extended SSDs such as Coins. 2) switching to defection for a fixed number of steps to neutralize a fixed debit threshold requires several manually selected parameters and does not generalize to complex environments. For the first problem, we use VI to assess the influence of one agent's action on another agent's expected return as a general measure of cooperativity. For the second, we develop the notion of an influence balance that is analogous to debit, but can be dynamically "paid off" via reciprocal influence at each timestep instead of an abrupt switch from cooperation to defection.
>
> -   Our work can also be seen as a "cumulative reciprocator" (CURE; Li et al., 2022), which has been shown to improve the stability of cooperation compared to memory-1 TFT strategies, especially in the presence of errors (e.g., a stochastic policy). CURE depends on countable instances of cooperation and defection, which are used to compute the defection difference statistic $d(k)$ - maintaining a history of cooperation can improve stability by allowing agents to tolerate "accidental" defection from mostly cooperative opponents. While this works in simple matrix games such as IPD, it is more difficult to directly classify individual actions as cooperative or defective in temporally-extended games such as Coins, inspiring our use of the modified VI metric to measure continuous changes to expected return as a quantification of cooperation/defection. Our influence balance is then a continuous analogue to $d(k)$.
>
> -   We hope that this context makes clear to the reviewer the contributions of each part **as generalized components within a single TFT-like strategy, rather than disparate mechanisms**. We would be happy to revise our manuscript to make this clearer.
>
> > With respect to the baselines, it seems to me that the type of "cooperation" in LOLA/MFOS is different from that defined by the authors. The authors of LOLA use "cooperation" as a term, but it is a different phenomenon from the emergence of cooperation of interest for the authors.
>
> -   We would like to ask the reviewer for further clarification on specific reasons why it seems to them that these two types of cooperation are different. However, we provide our case here as to why we believe they are very similar. As stated in the paper and in our responses to other reviews, our method performs a form of opponent shaping (as do MFOS and LOLA) by influencing opponent returns following particular actions via reciprocation, with the goal of shaping policy updates towards actions that are favorable w.r.t. the Reciprocator's own expected return.
>
> -   In social dilemmas, the environment's reward structure induces mutually defective behavior from naive learners, in which they myopically optimize for their own individual returns to converge to a Pareto-dominated outcome. Opponent shaping work such as LOLA, MFOS, and our work seek to take actions which guide opponents towards cooperative solutions, for the reason that **cooperative behavior from the opponent leads to increased return for the shaping agent** in SSDs. We distinguish this class of cooperation from a large body of work in MARL in which agent objectives are explicitly modified to include "altruistic" goals - which we once again stress fails to achieve cooperation and is exploited when faced with self-interested opponents.
>
> > With respect to the notation, I see what the authors mean, but I would suggest trying to improve/correct the notation in terms since those terms are not essentially defined in the formula, in my opinion.
>
> -   We appreciate the reviewer's attention to detail - as stated in our initial rebuttal, we have updated our draft to define the unclear terms (the counterfactual reward function) and fixed the summation term. Additionally, we will clarify that $\mathbf{a}^{-i}$ denotes the vector of joint actions excluding that of agent $i$.

---

> > ### Comment · Reviewer_SrMN · 2024-08-13
> >
> > Many thanks for the further clarifications.
> >
> > Just a quick note: by doing opponent shaping, you do not have emergence of cooperation, but cooperation by design in my opinion. Again, LOLA and MFOS are generic frameworks in themselves, not designed for the problem of emergence of cooperation in my opinion.

---

> ### Author Response · Authors · 2024-08-13
>
> We appreciate the reviewer's engagement in this process and address both our choice of baselines and our choice of verbiage for "emergent cooperation."
>
> Regarding our choice of baselines:
>
> - Because we position our work as an opponent shaping method with specific advantages over existing work (namely, improved sample efficiency, reliance only on first-order derivatives, and resistance to exploitation), we maintain that the baselines used in this paper such as LOLA and MFOS are appropriate. We are not aware of other shaping/cooperation-related baselines in the literature that are equally appropriate, keeping in mind that our method demonstrates cooperation in a setting of **simultaneous** learning in an **unmodified** environment.
>
>   -   Agents with modified prosocial objectives such as Empathic DQN (Bussmann et al., 2019), inequity aversion (Hughes et al., 2018), Altruistic Gradient Adjustment (Li et al., 2024) cannot induce cooperative behavior from self-interested opponents in SSDs and would therefore be trivial baselines.
>
>   -   Other opponent shaping methods such as LIO (Yang et al., 2020) and Learning to Penalize other Learning Agents (Schmid et al., 2021) explicitly modify the environment by expanding the action space to allow agents to directly influence the rewards of other agents, making them inapplicable to the original unmodified games.
>
>   -  The most similar works to ours are amTFT (Lerer & Peysakovich) and the hierarchical model of Kleiman-Weiner (2016) require a priori knowledge of cooperative and defective strategies and therefore cannot be extended to the simultaneous learning setting.
>
> - Finally, we note that shaping methods, unlike prosocial methods (as enumerated above), are typically evaluated head-to-head against a variety of other agent types. Our agent's resistance to exploitation and strong performance against higher-order shapers is a key result of this paper that would only be possible to demonstrate in these head-on comparisons against strong modern opponent-shaping baselines - the success of our method against prosocial agents would be trivial and of little interest to the community, in our opinion.
>
> Regarding our usage of the term "emergent cooperation":
>
> -   We appreciate the reviewer's perspective and concede that "emergent cooperation" may be an overloaded term in the context of opponent shaping, since the mutual cooperation is induced by the reward structure of SSDs **combined** with opponent shaping abilities rather than the latter alone. Therefore, we have toned down the language regarding this claim in our current draft.

---

### Official Review · Reviewer_wp16 · 2024-07-21

**Soundness:** 2
**Presentation:** 2
**Contribution:** 2
**Rating:** 6
**Confidence:** 3

**Summary:**

The authors introduce Reciprocators, RL agents that are intrinsically motivated to reciprocate the influence of other player's actions on the agent's returns. They show that this promotes cooperation in social dilemmas.

**Strengths:**

Originality:

- I do not believe this method has been proposed before

Quality:

- The results shown are strong. The fact that reciprocators are more resistant to extortion from MFOS is a previously unseen result.

- The authors get strong results on the coin game, a challenging sequential social dilemma.

Clarity:

- The paper is clearly written.

Significance:

- As learning models become more prevalent, algorithms that learn to cooperate and are aware of each other's learning becomes increasingly important and significant. This paper proposes one such algorithm for this setting.

**Weaknesses:**

This paper claims that it does not modify the reward structure of the environment. However, adding an intrinsic reward to cooperate seems like it very much does modify the reward of the agents. It's not particularly interesting that, when given reward for promoting cooperation, agents perform behaviors that promote cooperation. What makes prior work on opponent shaping interesting is that when given a *purely self-interested objective* (ie anticipating the opponent's learning updates), cooperation emerges naturally.

In particular, the highlighting of results in Table 1 is misleading: The goal of prior works in opponent shaping is not to simply promote cooperation, but to promote *rational* cooperation (ie without being exploitable). For example, in Table 1, if I were self-interested and were told to pick an algorithm, I should always pick MFOS since it dominates all other choices. I do believe that this difference in intention between this work and the prior literature / baselines should be highlighted, as direct comparisons do not seem appropriate.

Misc: The paper references the exponential sample complexity of M-FOS a few times. It might be good to reference the relevant paper [1] discussing that.

Fung, K., Zhang, Q., Lu, C., Wan, J., Willi, T., & Foerster, J. Analysing the Sample Complexity of Opponent Shaping.

**Questions:**

1. Have you considered trying your method on other SSD's beyond Coin Game? [1] discusses some of the downsides of using the coin game and introduces some new environments for doing so, available in [2] and [3]. I do not expect experiments to complete in time for the rebuttal, but it could be interesting for a camera-ready copy or future work.

2. For Line 212: doesn't augmenting coin game with the time remaining de-construct the social dilemma? At the very last time step, one should always defect. By induction, this then applies all the way back to the first time step. State aliasing prevents this from happening.

[1] Khan, Akbir, et al. "Scaling Opponent Shaping to High Dimensional Games."

[2] https://github.com/ucl-dark/pax

[3] Rutherford, Alexander, et al. "JaxMARL: Multi-Agent RL Environments and Algorithms in JAX." Proceedings of the 23rd International Conference on Autonomous Agents and Multiagent Systems. 2024.

**Limitations:**

The authors have addressed the limitations of their work.

---

> ### Author Rebuttal · Authors · 2024-08-06
>
> Weaknesses
>
> W1
>
> -   Although the central finding of this paper is that we are able to produce cooperative behavior through our intrinsic reward, we emphasize that the underlying mechanism of the reciprocal reward is one of opponent shaping rather than a prosocial inclination to cooperate. The intrinsic reward's purpose is to alter the opponent's expected returns following various actions, in order to shape their policy updates towards a desired behavior. For example, suppose that the opponent takes an action which marginally benefits itself but significantly harms the return of the Reciprocator. Then, the Reciprocator is motivated to punish this behavior by reducing the opponent's return following that action and therefore negating the action's advantage. Future learning updates should then lower the probability of this action under the opponent's policy.
>
> -   Concretely, this differentiates it from other methods which add an intrinsic reward for the sole purpose of cooperation such as the inequity aversion of Hughes et al., which are able to produce cooperative behaviors ONLY if the agent population is entirely composed of other agents with the modified cooperative reward. On the other hand, we show that our agent is able to induce cooperative behavior from purely self-interested agents, leading to higher returns in the long run.
>
> -   However, we acknowledge that this is a real concern in the symmetric game when Reciprocators go head-to-head. To address this, we recorded the cumulative reciprocal reward received in each episode in Appendix Figure B1 (incorrectly linked as A5 in the text) and provided an explanation in L 295 to emphasize that the intrinsic reward is small relative to the extrinsic reward and does not transform the overall reward structure of the game into a purely cooperative one.
>
> -   We will update our manuscript to clarify these points.
>
> W2
>
> -   Our original motivation for including this figure came from the key concerns laid out in both the MFOS and LOLA papers: specifically, the idea of an agent-order "arms race," in which agents of successively higher orders (e.g., meta-MFOS or nth-order LOLA) can readily exploit lower-order versions of themselves.
>
> -   We therefore positioned the reciprocal reward as an addition to basic first-order RL agents that could resist exploitation by higher-order agents without requiring meta-learning or higher-order derivatives, while NOT exploiting other naive learners itself, a phenomenon which we sought to demonstrate in the IPD-Rollout experiments as explained in section 6.1, L269.
>
> -   However, we agree that this is a very fair point - since MFOS and LOLA are specifically designed as opponent shaping algorithms rather than cooperation-inducing algorithms, they should not be evaluated in terms of their ability to achieve cooperation with different opponents. We have removed the bolding in the table and instead modified the caption and text to explain these advantages of the Reciprocator in our draft.
>
> W3
>
> -   An oversight on our part, thank you for pointing this out! We have added the relevant citations.
>
> Questions
>
> Q1
>
> -   To the best of our knowledge, previous works in the opponent shaping literature have conducted evaluations only on simple iterated matrix games and Coins, and so we focused on these settings. However, we note that Reciprocators capture both types of memory necessary to achieve shaping as identified in [1]: the counterfactual baselines trained on replay buffers serve as a way to capture inter-episode context, and the recurrent policies and influence balance capture intra-episode history. Therefore, we expect our method to handle the augmented SSDs introduced in [1]. We thank the reviewer for bringing this to our attention and will conduct experiments on "CoinGame in the matrix" for a camera-ready copy.
>
> Q2
>
> -   This is an interesting insight! We apologize for the lack of clarity here - we augment the observations input to the joint and counterfactual Q-function estimators as a way to stabilize the estimates used to compute the value influence (VI), but do not provide them as part of the input to the policy networks. We have updated the text of the paper to clarify this distinction.
>
> -   We note that recurrent policies are mandatory in order to reciprocate/shape the opponent as detailed in [1] and used (in the form of GRUs) in other shaping methods such as MFOS, POLA, LOQA, etc., and RNNs are by nature capable of tracking elapsed time during an episode regardless of a time-augmented observation. However, we see empirically that the presence of recurrence in NLs or Reciprocators does not prevent the induction of cooperation in ours or previous works.

---

> > ### Comment · Reviewer_wp16 · 2024-08-11
> >
> > W1/2:
> >
> > It would be good to see the clarifications mentioned in the updated manuscript. That being said, the authors' response has largely addressed my concerns, so I will raise my score.
> >
> > W3:
> >
> > There was a typo on my end, where I put [1] I was referring to this paper:
> >
> > Fung, K., Zhang, Q., Lu, C., Wan, J., Willi, T., & Foerster, J. Analysing the Sample Complexity of Opponent Shaping.

---

> > > ### Author Response · Authors · 2024-08-14
> > >
> > > We thank the reviewer for their consideration of our work and will be sure to include the clarifications in our future manuscript!

---

### Decision · Program_Chairs · 2024-09-25

**Decision:**

Accept (poster)

**Comment:**

This paper proposes a new co--player shaping method that is meant to be better for cooperatively resolving sequential social dilemmas. Most reviewers were happy with the experiments and baselines, as well as the connection to broader themes in the literature concerned with these questions.

Reviewer SrMN disagreed with the others and argued that the baselines were not appropriate. But the other reviewers disagreed with them. I looked through the paper myself also and I concur with the other reviewers. I do think MFOS is an appropriate baseline here, and I notice that the results achieved for it here look very closely comparable to those reported in the MFOS paper itself. So this looks fine to me.

I note also that the authors promised to include the discussion which emerged in the conversation around the reviews here of the relationship between this work and Jaques et al. (2019) "Social Influence...". Like the reviewer who asked about it, I was also wondering about the relationship between the two papers, and I thought your explanation was interesting. It's a very good idea to include it in the related work for the camera ready version.